# The Protective Effect of *Trichoderma asperellum* on Tomato Plants against *Fusarium*
*oxysporum* and *Botrytis cinerea* Diseases Involves Inhibition of Reactive Oxygen Species Production

**DOI:** 10.3390/ijms20082007

**Published:** 2019-04-24

**Authors:** Verónica I. Herrera-Téllez, Ana K. Cruz-Olmedo, Javier Plasencia, Marina Gavilanes-Ruíz, Oscar Arce-Cervantes, Sergio Hernández-León, Mariana Saucedo-García

**Affiliations:** 1Instituto de Ciencias Básicas e Ingeniería, Universidad Autónoma del Estado de Hidalgo, Pachuca-Tulancingo de Bravo Kilómetro 4.5, Mineral de la Reforma 42184, Hidalgo, Mexico; iran.poxi@gmail.com; 2Instituto Tecnológico de Acapulco, Carr. Cayaco Puerto Marqués s/n, Del PRI, Acapulco 39905, Guerrero, Mexico; karen_link_94@hotmail.com; 3Departamento de Bioquímica, Facultad de Química, Universidad Nacional Autónoma de México, Mexico City 04510, Mexico; javierp@unam.mx (J.P.); gavilan@unam.mx (M.G.-R.); 4Instituto de Ciencias Agropecuarias, Universidad Autónoma del Estado de Hidalgo, Avenida Universidad Km. 1, Rancho Universitario, Tulancingo-Santiago Tulantepec, Tulancingo 43600, Hidalgo, Mexico; oarce@uaeh.edu.mx (O.A.-C.); sergio_hernandez@uaeh.edu.mx (S.H.-L.)

**Keywords:** *Trichoderma asperellum*, reactive oxygen species, kaolin, *Fusarium oxysporum*, *Botrytis cinerea*

## Abstract

*Trichoderma* species are fungi widely employed as plant-growth-promoting agents and for biological control. Several commercial and laboratory-made solid formulations for mass production of *Trichoderma* have been reported. In this study, we evaluated a solid kaolin-based formulation to promote the absortion/retention of *Trichoderma asperellum* in the substrate for growing tomato plants. The unique implementation of this solid formulation resulted in an increased growth of the tomato plants, both in roots and shoots after 40 days of its application. Plants were challenged with two fungal pathogens, *Fusarium oxysporum* and *Botrytis cinerea*, and pretreatment with *T. asperellum* resulted in less severe wilting and stunting symptoms than non-treated plants. Treatment with *T. asperellum* formulation inhibited Reactive Oxygen Species (ROS) production in response to the pathogens in comparison to plants that were only challenged with both pathogens. These results suggest that decrease in ROS levels contribute to the protective effects exerted by *T. asperellum* in tomato.

## 1. Introduction

One goal of modern agriculture is the production of safe food through sustainable and eco-friendly practices. These involve a cutback in the use of chemicals in the fields to reduce potential environmental damage. In this regard, biological agents constitute an excellent alternative to replace chemicals for pest control or growth improvement. *Trichoderma* species are widely used for field application as biofungicides against pathogens such as *Botrytis cinerea*, *Fusarium* spp., *Pythium* spp., *Rhizoctonia solani* and *Sclerotium rolfsii* on crops of economic importance [1,2].

*Trichoderma* is a ubiquitous genus of filamentous fungi that growths in the rhizosphere and colonize plant roots as an opportunistic, avirulent plant symbiont [3]. *Trichoderma* spp. receives nutrients from root exudates in exchange for plant protection against biotic and abiotic stresses [4]. The interface between *Trichoderma* and plant is established by early events that include the release of elicitors from the cell walls from both organisms and the secretion of low molecular weight compounds, peptides and proteins from the fungus, leading to the recognition by the plant [5]. This first contact activates plant defense responses that promote disease resistance.

Several modes of action have been described to explain *Trichoderma* antagonism toward fungal pathogens and include mycoparasitism, antibiotic production and competition for nutrients [6]. Moreover, *Trichoderma* also exerts an indirect control against pathogens through the induced systemic response (ISR) in plant cells that results in an enhanced defense. The first evidence of induced resistance by *Trichoderma* was provided by Bigirimana et al. [7], who detected that soil inoculation with *T. harzianum* strain T-39 induced resistance to diseases caused by *B. cinerea* and *Colletotrichum lindemuthianum* in bean leaves. Systemic acquired resistance (SAR) and ISR are two forms of induced resistance, wherein plant defenses are preconditioned by prior infection or treatment that results in resistance against subsequent challenge by a pathogen or parasite [8]. ISR is activated mostly against necrotrophic pathogens, is mediated through the jasmonate (JA) and ethylene (ET) pathways, and goes without pathogenesis-related proteins (PRP) accumulation [9,10]. On the other hand, SAR is triggered by local infection and involves salicylic acid (SA) signaling pathway, requires PRP and is mostly effective against biotrophic and hemibiotrophic pathogens [11].

The effectiveness of microorganisms used as biocontrol agents depends on elements including environmental factors, survival in the soil, the formulation quality and the application protocol. Bacterial and fungal cells may be immobilized in solid carriers such as kaolin for preservation and protection from the external environment [12,13,14,15]. Kaolin is an inexpensive clay that essentially consists of minerals of kaolinite (ideal formula, Al_4_(Si_4_O_10_)(OH)_8_) group.). It is a unique industrial mineral because is chemically inert over a relatively wide pH range, is white, has good covering or shielding capacity when used as a pigment, coating film or filling material, and is soft and nonabrasive [16]. Kaolin has been used to carry and preserve fungi [12,14,15,17,18,19,20] and bacteria [20,21,22] with herbicide activities. In addition, kaolin has also been used in formulations to produce entomopathogenic nematodes [23].

Tomato is the highest top-ranked vegetable grown over the world. It accounts for more than 15% of global vegetable production (over 177 million metric tons in 2016; www.fao.org/faostat) [24]. Diseases are a major limiting factor for tomato production, and among them, fungal diseases are quite severe. *Fusarium oxysporum* is a soil-borne, hemibiotrophic, fungal pathogen that reduces the productivity of tomato crops in greenhouse and field. The fungal pathogen reaches the xylem vessels in the late stages of infection inducing progressive wilting and eventually plant death [25]. Chemical methods used to control soil-borne pathogens include biocide application; however, this practice is environmentally hazardous, and some chemicals have been phased out of use [26]. Host resistance and biocontrol agents are replacing these traditional pest management practices [27].

*B. cinerea* is a necrotrophic pathogen with a wide host-range that colonizes senescent or dead dicot plant tissues and fruits to produce gray mold. *Botrytis* species attack nursery plants, vegetables, ornamental, field and orchard crops, as well as stored and transported agricultural products [28]. Control of *B. cinerea* is difficult due of its several attack modes, diverse hosts and inoculum sources, and because it can survive as mycelia and/or conidia for extended periods as sclerotia in crop debris. Several fungicides have been employed to control *Botrytis* diseases. However, resistance to cyprodinil and fludioxonil occurs in strains of this species [29]. Moreover, the current cost of releasing a new fungicide or biological control agent to the market is so high that only major crops attract sufficient interest by agribusiness [30].

A *T. harzianum* formulation based on solid bentonite-vermiculite is effective for controlling *Fusarium* wilt and promoting melon growth in greenhouse conditions [31]. This formulation has also been used in tomato plants [32]. In this work, we prepared a *T. asperellum* formulation using kaolin instead bentonite and analyzed its efficiency in controlling *F. oxysporum* or *B. cinerea* infections in tomato plants. Such formulation notably reduced disease progression against both fungal pathogens. Moreover, we found that *T. asperellum* prevented ROS generation. These low ROS levels were associated with disease control promoted by *T. asperellum*.

## 2. Results

### 2.1. Kaolin Formulation Allows Trichoderma Asperellum High Biomass, Viability and Stability

*Trichoderma* produces large amounts of fungal spores and this characteristic makes it ideal for inoculum production under laboratory conditions. In this study, we used a solid formulation devised to promote *Trichoderma* proliferation and survival in soil. The solid formulation consisted of an oat flakes/kaolin-vermiculite mixture that functions as an inexpensive, simple and readily available source of nutrients.

To evaluate *T. asperellum* spore viability in the growing substrate, CFUs were quantified several days after application. As shown in Figure 1, approximately 1 × 10^7^ CFU/g soil remained at 4 dpi, and those levels were maintained throughout the experiment (40 days), indicating an optimal retention and survival of *T. asperellum* in the solid matrix. Moreover, this formulation showed a ten-fold higher biomass production of *Trichoderma* inoculum than the bentonite-vermiculite formulation employed with *T. harzianum* [31]. A possible explanation for this logarithmical difference may be due to the formulation pH. According to Singh [33], *T. asperellum* produces high biomass at pH 7.0 with a slow preference at pH 5.5. In our study, the pH recorded in the kaolin was 5.9, suggesting that an acidic milieu positively influences spore production.

### 2.2. Trichoderma Asperellum Stimulates Plant Growth on Tomato Plants

To characterize the effect of *T. asperellum* on plant growth and development, two-week-old tomato seedlings were grown in sterile soil or in *T. asperellum*-treated soil mixes. After 40 days, growth parameters were measured. Plant absolute growth rate and chlorophyll content were 22% and 16% higher, respectively in *T. asperellum*-treated plants than in control plants (Table 1). Both fresh and dry weight were 30% higher in treated plants, as well. Leaf area was not significantly different between the two treatments (Table 1).

### 2.3. Trichoderma Asperellum Reduces Fusarium Oxysporum Wilt Symptoms in Tomato Plants

The effectiveness of *T. asperellum* as a biocontrol agent was evaluated by recording the disease symptoms induced by *F. oxysporum* (Figure 2). Two-week-old plants were planted in the *T. asperellum-*solid formulation or mock, and three weeks after were inoculated with *F. oxysporum*. The plants were grown for three weeks post-inoculation (wpi) before disease symptoms assessment. Less severe disease symptoms caused by *F. oxysporum* were recorded in *T. asperellum*-inoculated plants than in untreated plants. Symptoms included wilting and stunting (Figure 2A). This result was concordant with the reduction in fresh-weight loss triggered by fungal pathogen infection in *T. asperellum*-inoculated plants in reference to non-pretreated plants (Figure 2B).

Disease severity was estimated according to leaf damage established by the scale shown in Figure 2C (upper panel) for plants after one and three weeks of infection with *Fusarium*. While plants infected with *F. oxysporum* mainly exhibited the most severe symptoms, after 3 weeks, *T. asperellum*-treated plants and infected with *F. oxysporum* displayed disease severity values averaging between one and two. These results support the role of *T. asperellum* as a successful biocontrol agent against *Fusarium* wilt.

*F. oxysporum* plant colonization was analyzed by cutting stem sections from different nodes of non- and pretreated plants and plated on PDA [34].

As it is shown in Figure 2D, *F. oxysporum* colonized cotyledon node and first node of mock plants. In a similar way, *F. oxysporum* was detected in the same nodes of *T. asperellum*-pretreated plants. However, the biocontrol agent showed faster growth. This result contrasts with the report by Martínez-Medina et al. [32] who did not detect *Trichoderma* presence in the shoots of tomato plants inoculated in the roots. In this regard, a possible contamination cannot be ruled out but neither that *T. asperellum* can be translocated through the vascular system.

### 2.4. Trichoderma Asperellum Induces Systemic Resistance against Botrytis Cinerea

To investigate the protective systemic effects of *T. asperellum* against fungal disease, detached leaves from mock and pretreated *T. asperellum* plants were inoculated with *B. cinerea*. Lesion development was recorded two weeks after infection. Detached leaves from *T. asperellum*-pretreated plants showed lower and less severe lesions compared to non-treated plants (Figure 3A,B). These results are in accordance with Martínez-Medina [32], who found a 30% decrease in lesion size in plants inoculated with *T. harzianum* and infected by *B. cinerea*.

### 2.5. Trichoderma Asperellum Supresses ROS Production against Necrotrophic Pathogens

ROS production is an early event during plant defense responses triggered by pathogens attack. In this context, the effect of *T. asperellum* on ROS accumulation was investigated in tomato leaves infected with *F. oxysporum* and *B. cinerea* in the first 24 h (Figure 4A,B, respectively). ROS production was recorded three weeks after inoculation of tomato plants with *T. asperellum*.

As Figure 4 shows, ROS production was markedly diminished in tomato plants preinoculated with *T. asperellum* in comparison to non-treated plants infected with both fungal pathogens. ROS production was detected at 3 hpi, and it was sustained until 24 h and 12 h in plants infected with *F. oxysporum* and *B. cinerea*, respectively.

## 3. Discussion

We found that an oat/kaolin-vermiculite-based formulation for *T. asperellum* application was highly effective for sustaining the biocontrol agent and to protect tomato plants from *F. oxysporum* and *B. cinerea*. This protective effect was associated with the reduction of ROS production in response to pathogen infection. Under these conditions, where grinding or extrusion of bioformulation were not considered, *T. asperellum* viable biomass, determined as CFUs, remained unchanged over 40 days and no contamination was detected in the culture examinations, indicating that kaolin was not a source of contaminants, as it was previously reported [12]. Moreover, *T. asperellum* retained its physiological and biochemical activities in the solid formulation since its application to tomato plants led to enhanced growth and tolerance against pathogens.

Many reports have shown that *Trichoderma* spp. improves the growth in many plant species [11]. In this study, we demonstrated that *T. asperellum* improved plant growth parameters including plant absolute growth rate, and fresh and dry weight with a significant difference relative to mock treatment, except leaf size. It is important to highlight the enhancement of shoot and root growth as it has been reported [35,36].

Chlorophyll content was also increased in *T. asperellum*-treated plants as was observed in melon and cacao plants inoculated with *T. harzianum* and *T. asperellum*, respectively [31,37]. This result suggests an optimal physiological status of plants.

Besides its plant growth stimulatory activity, *Trichoderma* spp. are fungi that reduce negative effects by plant pathogens. In our study, we investigated the suppressive role of *T. asperellum* against two fungal pathogens of tomato plants. *T. asperellum* reduced disease symptoms development upon infection or *F. oxysporum* or *B. cinerea* in tomato plants.

It has recently been reported that SA and JA are involved in *T. virens-*mediated resistance in tomato against *F. oxysporum*. Jasmonic acid-deficient *def-1* and salicylic acid-deficient *NahG* mutants are more susceptible to infection. Measurement of JA in wild type plants revealed that *T. virens* cultured on barley grains or the combination *T. virens* plus *F. oxysporum* showed higher JA levels and lower disease incidence. By comparison, *F. oxysporum*-treated plants only showed a small increase in the JA levels [38]. These results suggest that *T. virens* is the responsible agent to induce JA changes in the plant like a defense mechanism. In contrast, SA content increased in response to the pathogen. However, SA was only slightly built up in plants treated with *T. virens* cultured on barley grains or in the combination of *T. virens* plus *F. oxysporum* [38]. The involvement of JA and SA has also been demonstrated in the resistance mediated by *T. harzianum* of tomato plants against *B. cinerea*. The resistance promoted by *T. harzianum* is also lost in *def-1* and *NahG* mutants. In addition, *def-1* mutants showed higher colonization of *B. cinerea* in comparison of wild type infected plants [32], suggesting that JA is controlling the pathogen proliferation in the plant tissues.

The protection mediated by *T. asperellum* against both pathogenic infections could be related to growth promotion, at least in part. However, in this study we report that *T. asperellum* inhibited ROS the common production upon pathogen infection. To avoid oxidative stress, the cells use a radical scavenging mechanism to neutralize free radicals or reactive species. Superoxide dismutase (SOD) is an enzyme whose activity is involved in the earliest defense responses. It is considered the first detoxifying line. SOD catalyzes the dismutation of superoxide anion to hydrogen peroxide [39]. *T. harzianum* has been shown to increase SOD activity in tomato plants after *F. oxysporum* infection [40]. Nevertheless, we only detected a weak H_2_O_2_ accumulation in the tomato leaves of plants pretreated with *T.asperellum*, which were subsequently pathogen-infected with reference to unpretreated plants challenged with the pathogen. According to Zehra et al. [40], the pretreatment of tomato plants with *T. harzianum* supresses ROS builds up by enhancing mechanisms such as antioxidant defense mediated by catalase and ascorbate peroxidase activities in response to *F. oxysporum* infection.

*B. cinerea* secretes toxins to kill plant cells. The most known are botrydial and its by-products and botcinic acid derivatives [41,42]. Besides these toxins, *B. cinerea* itself produces ROS in the plant during the infection process as a virulence factor. H_2_O_2_ accumulates in the early steps of infection, both in germinating spores and in the infection cushions [43]. *B. cinerea* shows resistance against the oxidative burst induced as a plant-host defense response, protecting itself with an extracellular catalase activity [44]. Beyond survival to an oxidative environment during plant–pathogen interaction, *B. cinerea* also stimulates ROS production in the plant. Early production of ROS and host cell death are indicative of a successful infection by *B. cinerea* [45]. In some plant–pathogen interactions, PCD promotes pathogen growth, especially those pathogens that secret toxins to the host cells [46]. ROS increase in Arabidopsis enhances necrosis induced by *B. cinerea* and this necrosis is dependent on host-activated HR and not by fungus toxicity, since *dnd1*, a mutant with HR-deficiency is asymptomatic in response to *B. cinerea* infection [47]. Moreover, *B. cinerea* growth is fostered by ROS generation and HR establishment. The oxidative burst is effective to contend with biotrophic pathogens, but it is unable to protect the host against necrotrophs [48]. The above results demonstrated a correlation between ROS and lesion formation in response to *B. cinerea* infection in tomato plants, as was previously reported in Arabidopsis [47]. Massive necrosis was observed in plants infected with *B. cinerea* without *T. asperellum* pre-treatment. Under this condition, *Botrytis* lesion formation increased two-fold as compared to plants pretreated with *T. asperellum*. Regarding H_2_O_2_ production, this was suppressed at all times as evaluated by *T. asperellum* pretreatment suggesting that the biocontrol agent promoted resistance by inhibiting ROS formation and HR-like lesions.

A possible explanation for the mechanism by which *Trichoderma* spp mediates resistance in response to *F. oxysporum* and *B. cinerea* is through JA involvement, since this hormone controls fungi proliferation, up-regulates the expression of defense genes, induces the activities of catalase and ascorbate peroxidase and shows minimal lipid peroxidation and cell death after a challenge infection with these pathogens. Further research is required to elucidate the JA participation in the resistance induced by *Trichoderma* spp.

## 4. Materials and Methods

### 4.1. Plant Material

*Lycopersicon esculentum* genotype Vita was used in the experiments. Seeds were surface-sterilized in 0.5% sodium hypoclorite for 5 min, and rinsed twice with sterile water for 10 min before use. Seeds were germinated on 200-cavity trays with sterile peat moss and perlite (agrolite). Seedlings were grown in a greenhouse under 14 h light and 10 dark photoperiod at 27 °C and 12 °C, respectively.

### 4.2. Trichoderma Inoculum Preparation

The bio-agent was obtained from Dr. Hernández at Centro de Biotecnología Genómica (IPN), México. A pure culture of *T. asperellum* was maintained on PDA plates at 25 °C for 7 days. For inoculum preparation, a solid formulation (20 g of oat, 50 mL of kaolin, 100 mL of vermiculite and 60 mL of water (modified from Martínez-Medina et al. [31]) was sterilized and the substrate was later inoculated with *T. asperellum* (1 × 10^5^ spores mL^−1^) under sterile conditions and incubated for one week at 28 °C.

### 4.3. Inoculation of Tomato Plants with T. Asperellum

After complete plant emergence at 2 weeks, individual seedlings were thinned out to 6-inch pots containing commercial peat moss and vermiculite (7:3, *v*/*v*) supplemented with a mixture containing the solid substrate previously inoculated with *T. asperellum* or mock. Plants were returned to the green house.

### 4.4. Quantification of T. Asperellum in Growing Media

To quantify *T. asperellum* in the growing media, one gram of soil mixture containing the solid substrate inoculated with *T. asperellum* or mock was plated on potato dextrose agar (PDA) supplemented with 50 mg L^−1^ rose bengal and 10% streptomycin sulphate [31]. Medium was pH 4.9 adjusted with lactic acid. Plates were incubated at 28 °C for 5 days and then colony-forming units (CFUs) were counted. Data were expressed as Log CFU per gram of dry substrate.

### 4.5. Evaluation of Tomato Plant in Growing Media Inoculated with T. Asperellum

Plant Absolute Growth Rate (AGR) was evaluated forty days after inoculation with. *T asperellum* by recording shoot height. AGR was calculated using the formula AGR = (H2 − H1)/(t2 − t1) [49]. Leaf area (cm^2^), chlorophyll content, fresh weight (g) and dry weight (g) were also evaluated at 40 days post inoculation. For leaf area, individual leaves of tomato plants were scanned and subsequently analyzed by Image J 1.44 [50]. Optical measurements were made on leaves of 10 plants using a SPAD-502Plus (Konica Minolta Sensing, Inc., Tokyo, Japan). Fresh and dry weight were measured with a precision analytical balance. All measurements were recorded for at least 10 plants per treatment with three independent repetitions.

### 4.6. Isolation and Identification of Fungal Pathogens

A *Fusarium* strain was isolated from rotted tomato (cv. Saladette) roots by placing the disinfected tissue on PDA plates and incubating for 7 days. The mycelia emerging from the tissue was transferred to a fresh PDA plate and a mycelial plug from this plate was resuspended in sterile water. Serial dilutions from these spore suspensions were performed and a 100-μL aliquot of a 100-fold dilution was plated on 1% agar and incubated for 24–48 h. Single-conidia germinating were transferred to PDA plates and an agar plug from these cultures was used to inoculate GYAM liquid media. The mycelial mat from 5-day-old culture was used to extract genomic DNA. The PCR reaction was performed for 30 cycles using the ef1 (forward primer: 5′-ATGGGTAAGGA(A/G)GACAAGAC-3′) and ef2 (reverse primer: 5′-GGA(G/A)GTACCAGT(G/C)ATCATGTT-3′) with an annealing temperature of 53 °C. The 700-bp amplicon was gel-purified and used for DNA sequencing with the ef22 internal reverse primer (5′-AGGAACCCTTACCGAGCTC-3′), according to Geiser et al. [51]. The 350-bp sequence was analyzed through the *Fusarium* ID platform (http://isolate.fusariumdb.org/) for species identification.

*B. cinerea* was isolated on PDA media from tomato fruits showing typical signs of grey mould. Plates were incubated at 22 °C with 14 h darkness and 10 h light for 7 days. Pathogenicity tests were performed following Koch’s postulates.

### 4.7. Evaluation of Effectiveness of T. asperellum against Fungal Pathogens in Tomato Plants

Roots of plants grown for 3 weeks in media with and without *T. asperellum* formulation were cut about 1 cm from the tips, and the roots were placed during 5 min in the *F. oxysporum* suspension (1 × 10^5^ conidia mL^−1^). Control plants were mock-inoculated with sterile distilled water. Disease progression was followed through photograph records of symptoms 3 weeks post-inoculation (wpi). Plant fresh weight was recorded at 1 and 3 wpi.

Disease severity was calculated by the number of leaves showing different levels of wilt symptom and expressed as percentage in reference of total number of leaves per plant. Symptoms were recorded at 1 and 3 wpi. Phenotypic analysis, disease severity and loss weight assays were repeated three times with 10 replicates per treatment.

To explore disease progression, a colonization assay was performed in non- and pretreated plants with *T. asperellum* for 3 weeks and infected with *F. oxysporum* for 3 weeks.

Stem sections were collected from cotyledon node, first and second nodes. Sections were surface sterilized in 70% ethanol and rinsed in sterile distilled water and placed on PDA supplemented with 200 mg/mL streptomycin at 25 °C. Photographs were taken after 5 days of incubation on PDA. The experiment was repeated three times.

The *B. cinerea* strain was routinely cultivated in Petri dishes containing PDA. For inoculation, detached leaves from 5-week-old plants were laid on Petri dishes containing two blotting filter papers (Whatman) wetted with sterile water, then spotted with 5-mm-diameter agar plugs containing growing hyphae from *B. cinerea* [52]. Lesions were measured using the ImageJ software employing a calibration scale [50] at 1 and 2 wpi. Experiments were repeated three times with 10 replicates per treatment.

### 4.8. Evaluation of T. asperellum Effect on H_2_O_2_ Production in Tomato Fungal-Infected Plants

DAB is taken up by living plant tissue and can be used to show H_2_O_2_ production when peroxidase activity is present [53]. Detection of hydrogen peroxide was conducted using 3,3′-diaminobenzidine (DAB) from Sigma-Aldrich. Leaves were infected with *F. oxysporum* or *B. cinerea*, respectively and collected at different times. Plant tissues were immersed with 1 mL of DAB liquid buffer solution for 2 h. After staining, the tissues were fixed in ethanol:glycerol:acetic acid 3:1:1 (*v*:*v*:*v*) (bleaching solution) placed in a water bath at 95 °C for 15 min. Leaves were reimmersed in bleaching solution until chlorophyll was completely depleted.

### 4.9. Statistical Analyses

Data on AGR, leaf areas, SPAD units, fresh and dry weight and disease severity were subjected to a pairwise comparison with Student’s *t*-test (*p* < 0.05) implemented in Excel software.

## 5. Conclusions

This study demonstrates the successful use of a solid formulation based in kaolin as inoculum with *T. asperellum* to enhance tomato plants growth and defense against pathogens. The fact that *T. asperellum* inhibits ROS accumulation makes it an excellent candidate to control fungal pathogens that promote oxidative stress as mechanism for successful disease establishment. The application of this bioformulation can have a positive impact on disease management in sustainable agriculture. The effective application of the bioformulation in the field requires more research on issues such as shelf life of the product and its viability during storage.

## Figures and Tables

**Figure 1 ijms-20-02007-f001:**
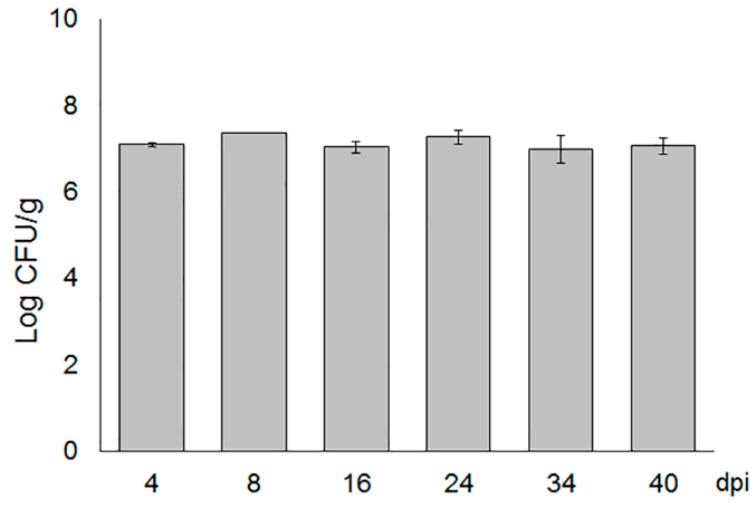
*Trichoderma asperellum* viability in the growing media. *T. asperellum* CFU per gram was quantified at various times after mixing with the growing substrate. The population was expressed as Log CFU/gram- Bars indicated SD from three independent experiments.

**Figure 2 ijms-20-02007-f002:**
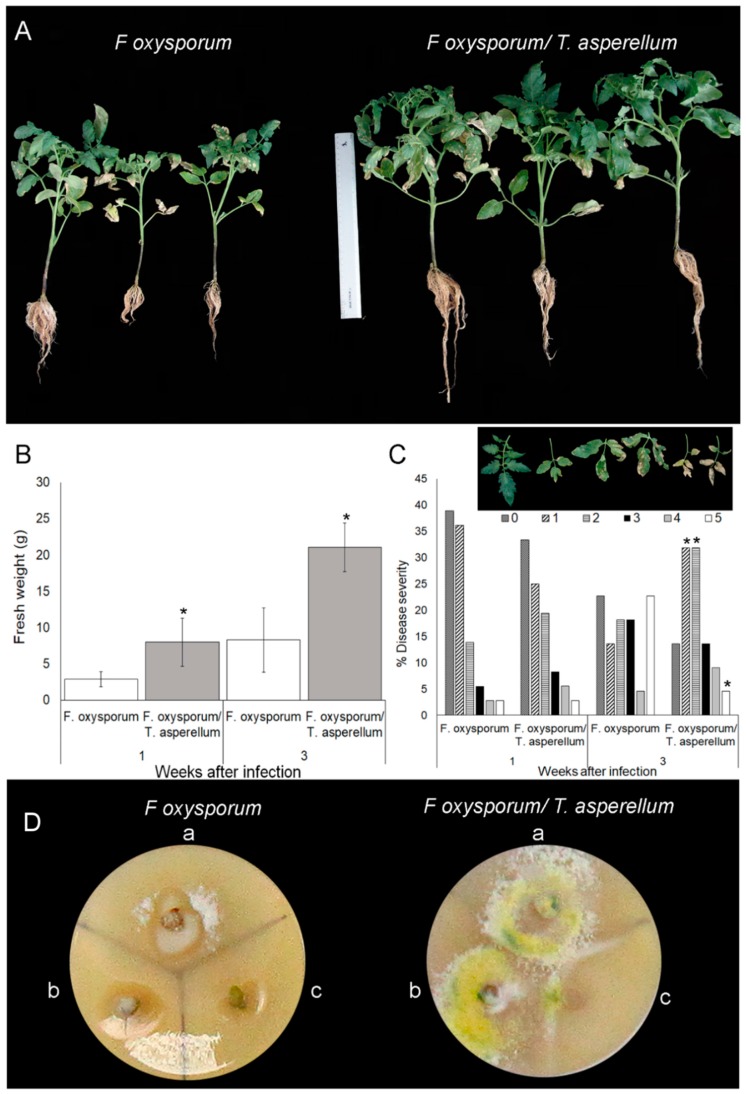
*Trichoderma asperellum* reduces *Fusarium* severity on tomato plants. (**A**) *Fusarium oxysporum* wilt and stunting symptoms on *T. asperellum* non- and preinoculated tomato plants after 3 weeks post infection. (**B**) Fresh weight was recorded at 1 and 3 weeks post-inoculation. (**C**) Disease severity (%) scores represent the number of leaves showing different levels of wilt symptoms shown in the upper panel in reference of total number of leaves of each plant. Symptoms were recorded at 1 and 3 weeks after infection. (**D**) Stem section were cut from cotyledon node (**a**), second node (**b**) and third node (**c**) of individual tomato plants non- and preinoculated with *T. asperellum* and infected after 3 weeks. Photographs were taken 5 days after incubation of nodes on PDA. Asterisks indicate significant differences among treatments at a given time (*p* ≤ 0.05), error bars indicate SEM.

**Figure 3 ijms-20-02007-f003:**
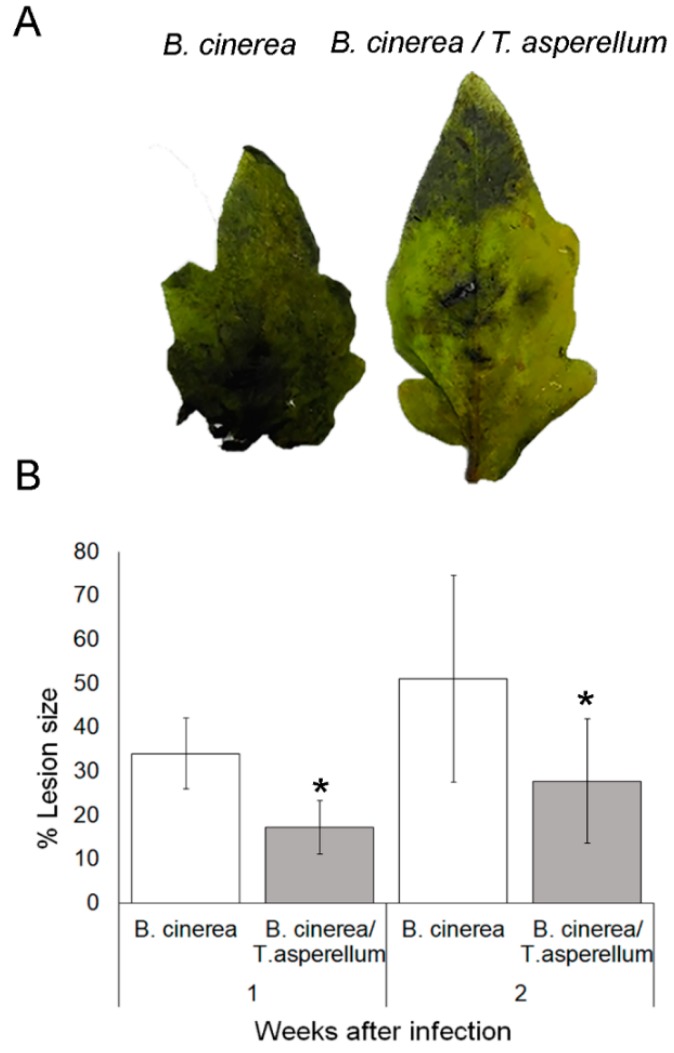
*Trichoderma asperellum* reduces *Botrytis* severity on tomato plants. (**A**) Detached leaves from non- and inoculated plants with *T. asperellum* were infected with 5 mm-diameter plugs of *B. cinerea*. Photographs were taken two weeks after infection. (**B**) Lesions were measured using Image J software 1 and 2 weeks after *Botrytis* infection. Asterisks indicate significant differences among treatments at a given time (*p* ≤ 0.05), error bars indicate SEM.

**Figure 4 ijms-20-02007-f004:**
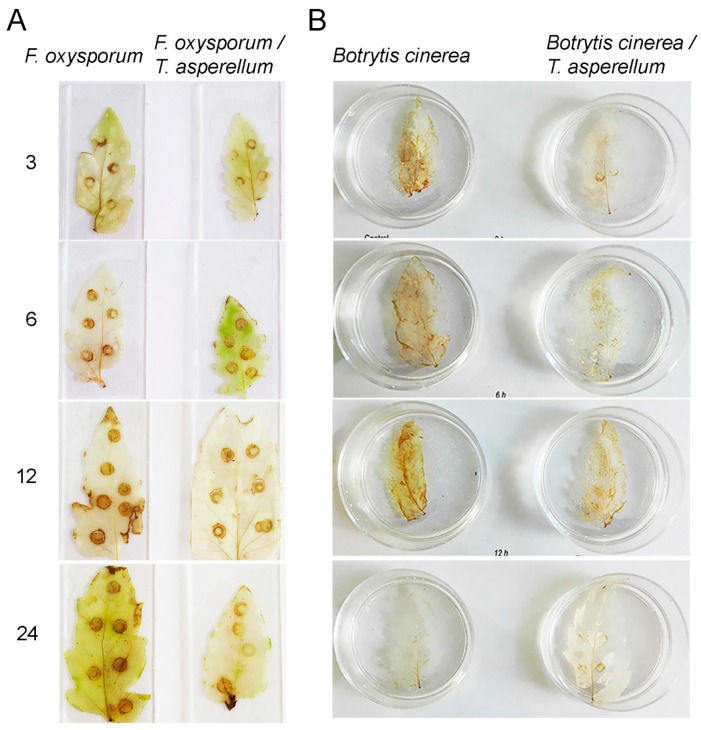
*T. asperellum* prevents ROS accumulation in response to *Fusarium* and *Botrytis* infection on tomato plants. Non- and pretreated tomato plants with *T. asperellum* were inoculated with *F. oxysporum* (**A**) or *B. cinerea* (**B**), and leaves were stained at different time periods (3, 6, 12 and 24 h) with DAB. Infection with *Fusarium* was in full plants while infection of *Botrytis* was in detached leaves. The experiments were repeated twice.

**Table 1 ijms-20-02007-t001:** Effect of *T. asperellum* on plant growth of tomato plants. Absolute growth rate, leaf area, SPAD units, fresh and dry weight of tomato plants after 40 days of *T. asperellum* inoculation are shown.

Treatments	AGR * (cm/day)	Leaf Area (cm^2^)	SPAD Units	Fresh Weight (g)	Dry Weight (g)
Control	0.394 ± 0.05 ^a^	10.91 ± 3.00 ^a^	22.46 ± 1.84 ^a^	12.72 ± 1.38 ^a^	0.957 ± 0.01 ^a^
*T. asperellum*	0.50 ± 0.08 ^b^	12.20 ± 2.90 ^a^	26.75 ± 2.09 ^b^	18.83 ± 0.58 ^b^	1.38 ± 0.01 ^b^

Data are means ± S.D. from ten biological replicates. Same letter in the column denote no significant difference between both treatments according a pairwise comparison using Student’s *t*-test (*p* < 0.05). * Based on plant height.

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
