# Peer review of "The Protective Effect of Trichoderma asperellum on Tomato Plants against Fusarium oxysporum and Botrytis cinerea Diseases Involves Inhibition of Reactive Oxygen Species Production"

_ijms, 2019, doi:10.3390/ijms20082007_

Round 1
Reviewer 1 Report
This paper is devoted to the use of Trichoderma asperellum as biocontrol agent for tomato plants against such fungal pathogens as F. oxysporum and B. cinerea. The authors evaluated an original kaolin-based formulation to promote the absortion/retention of T. asperellum in the substate for growing tomato plants. They also showed the inhibition of ROS production in response to the pathogens in comparison to the non-treated tomato plants. The work is done at a high scientific level, the results are reliable, the methods are described in detail and the experiments performed can be reproduced. The text of the article corresponds to the standards of scientific publication. I don´t have any doubt that this paper should be accepted without modifications.
Author Response
We are very grateful for reviewer 1 to examine the paper and to share with us the comments and observations around it according to his/her expertise.
English language was reviewed to attend some mistakes detected in the text. All them are remarked with track changes function.
In the word you can see the modifications performed according to the reviewer 2
Reviewer 2 Report
i. Subheading 4.6 . Fusarium oxysporum isolation and identification and 4.8 Botrytis cinerea isolation should be merge together as one heading of “Isolation and Identification of Pathogens”.
ii. What method applied for the identification of Botrytis cinerea
iii. Subheading 4.7 Evaluation of effectiveness of T. asperellum against B. cinerea in tomato plants
iii.and 4.9 Evaluation of effectiveness of T. asperellum against Fusarium oxysporum in tomato plants merge together.
iv. Statistical analysis: name the software used for analysis.
v. Figures clearly indicates the effects of T. asperellum on disease severity.
vi. Figure 1 A: Author can add plant no with description.
Author Response
We are very grateful for reviewer 2 to examine the paper and to share with us the comments and observations around it according to his/her expertise.
English language was reviewed to attend some mistakes detected in the text. All them are remarked with track changes function.
In Materials and methods section, the 4.6 and 4.8 subheadings were merged as well as 4.7 and 4.9, respectively.
The name of the software employed for statistical analyses was included (363 line).
Regarding to the B. cinerea identification is important to be noted that we did not use molecular barcodes. However, the identification of B. cinerea was based on morphological characteristics on PDA. We studied sclerotial characteristics (size, shape and arrangement on agar media), conidial characteristics (size, shape and ornamentation) and pathogenicity in tomato fruits.
Botrytis has been divided into two phylogenetically separated clades. Clade 1 that includes B. cinerea and three other species which infect only dicotyledonous plants and Clade 2 with 23 host-specific Botrytis species that infect predominantly monocotyledonous plants (Hyde et al., 2014; Fungal Diversity (2014) 67:21–125 DOI 10.1007/s13225-014-0298-1). As of yet, the only Botrytis specie reported to tomato plants is B. cinerea and this is responsable of gray mould. Due to the above points we did not use molecular analysis but we certainly recognize that DNA information is crucial to organisms identification.
About Figure 1A, we are doubtful about the comment. We are interested in knowing if it is necesary to perform some modification to the Figure 1 or 2A.